# Evaluation of Junk Food Consumption and the Risk Related to Consumer Health among the Romanian Population

**DOI:** 10.3390/nu15163591

**Published:** 2023-08-16

**Authors:** Magdalena Mititelu, Carmen-Nicoleta Oancea, Sorinel Marius Neacșu, Adina Magdalena Musuc, Theodora Claudia Gheonea, Tiberius Iustinian Stanciu, Ion Rogoveanu, Fallah Hashemi, Gabriela Stanciu, Corina-Bianca Ioniță-Mîndrican, Ionela Belu, Nicoleta Măru, Gabriel Olteanu, Alexandru-Tiberiu Cîrțu, Iuliana Stoicescu, Carmen Elena Lupu

**Affiliations:** 1Department of Clinical Laboratory and Food Safety, Faculty of Pharmacy, “Carol Davila” University of Medicine and Pharmacy, 3-6, Traian Vuia Street, Sector 2, 020956 Bucharest, Romania; magdalena.mititelu@umfcd.ro (M.M.); gabriel.olteanu@mst.umfcd.ro (G.O.); alexandru-tiberiu.cirtu@mst.umfcd.ro (A.-T.C.); 2Department of Biochemistry, Faculty of Medicine, University of Medicine and Pharmacy from Craiova, 200345 Craiova, Romania; carmen.oancea@umfcv.ro; 3Department of Pharmaceutical Technology and Bio-Pharmacy, Faculty of Pharmacy, Carol Davila University of Medicine and Pharmacy, 020945 Bucharest, Romania; neacsusorinelmarius@gmail.com; 4“Ilie Murgulescu” Institute of Physical Chemistry, 060021 Bucharest, Romania; amusuc@icf.ro; 5Center for IBD Patients, Faculty of Medicine, University of Medicine and Pharmacy from Craiova, 200345 Craiova, Romania; 6Press Office, Ovidius University of Constanța, 900527 Constanța, Romania; 7Department of Environmental Health Engineering, School of Health, Shiraz University of Medical Sciences, Shiraz 71468-64685, Iran; info.foo@gmail.com; 8Department of Chemistry and Chemical Engineering, Ovidius University of Constanta, 900527 Constanta, Romania; gstanciu@univ-ovidius.ro; 9Department of Toxicology, Faculty of Pharmacy, “Carol Davila” University of Medicine and Pharmacy, 020945 Bucharest, Romania; corina-bianca.ionita-mindrican@drd.umfcd.ro; 10Department of Pharmaceutical Technology, Faculty of Pharmacy, University of Medicine and Pharmacy of Craiova, 200638 Craiova, Romania; ionela.belu@umfcv.ro; 11Department of Anatomy, Faculty of Dental Medicine, “Carol Davila” University of Medicine and Pharmacy, 3-6, Traian Vuia Street, Sector 2, 020945 Bucharest, Romania; nicoleta.maru@umfcd.ro; 12Department of Chemistry and Quality Control of Drugs, Faculty of Pharmacy, “Ovidius” University of Constanta, 900001 Constanta, Romania; iuliana.stoicescu@univ-ovidius.ro; 13Department of Mathematics and Informatics, Faculty of Pharmacy, “Ovidius” University of Constanta, 900001 Constanta, Romania; clupu@univ-ovidius.ro

**Keywords:** healthy diet, fast food, ultra-processed foods, sweetened drinks, metabolic diseases, public health, food safety

## Abstract

Premature aging and degradative processes are mainly generated by unhealthy habits and an unbalanced diet. Quality of food and lifestyle are important factors in sano-genesis. Many imbalances and ailments have their origin in the adoption of an unbalanced diet and a disordered lifestyle. With the help of a transversal study carried out on the basis of a questionnaire, the consumption of junk food products among the population of Romania was evaluated; at the same time, an evaluation of the characteristics of the associated diet, as well as a series of lifestyle components (quality of rest, physical activity, evaluation of the state of health) was carried out. The data collected and processed indicate an increased tendency to consume junk food products in the 18–23 age group, and especially among obese respondents. Female respondents show a lower tendency toward an increased consumption of junk food products (OR = 0.703, 95% CI)—0.19–0.95, *p* = 0.011) compared to male respondents. The most consumed junk food products are fried potatoes (46.2%) and pastries (41.4%). Junk food products that show an increased tendency toward consumption addiction are fried potatoes (13.8%), sweets (12.4%), pastry products (11.1%), and sweetened drinks (11.2%). The poor quality of food from a nutritional point of view, and reduced physical activity, are reflected in the varied range of problems faced by the respondents: states of fatigue (62.4%), nervousness (37.5%), depression, anxiety, emotional eating, etc.

## 1. Introduction

The World Health Organization (WHO) draws attention to the danger of the excessive consumption of unhealthy foods and drinks, which represents an important risk factor for non-communicable diseases (NCDs) [1]. Junk food and sweetened and alcoholic drinks are included in the category of foods with a major risk for metabolic syndrome, which is complicated with serious consequences for health, and even the risk of premature death in the case of long-term excessive consumption associated with an unbalanced and sedentary lifestyle. These food products are characterized by a hypercaloric content, and being low in nutrients and rich in additives [2,3,4,5,6].

According to the country profile regarding the health of the Romanian population, published in 2021, and based on national statistics provided by Eurostat and OECD (Organization for Economic Cooperation and Development), the life expectancy in Romania is among the lowest in the European Union (EU), with cardiovascular diseases representing the main causes of death. Unhealthy behavior accounts for nearly half of all deaths, and the rate of reported alcohol consumption and unhealthy eating among the population is above the European average. The mortality from treatable causes is double the European average, and the avoidable mortality ranks third in the EU, being mainly caused by cardiovascular diseases, lung cancer, and excessive alcohol consumption [7].

The main causes of Romanian deaths are represented by behavioral and environmental risk factors, especially an unhealthy diet (25%), excessive tobacco consumption (17%), excessive alcohol consumption (7%), a sedentary lifestyle (2%), and pollution (7%) [7]. Numerous studies indicate the presence of pollutants in high concentrations in certain food products, causing serious alarm bells related to their safety in consumption [8,9]. Unhealthy eating is characterized by a low consumption of vegetables and fruits, and a high consumption of sugar and salt. This causes a reduced consumption of fibers, antioxidants, vitamins and minerals, which are valuable nutrients in the optimal functioning of the body [10,11]. The rate of obesity among Romanian children has also increased recently; according to 2018 statistics, one in five children aged 15 is obese [7].

Nutrition is of particular importance in maintaining good health, in the correct and harmonious development of the body, and in the prevention of certain ailments, and nutritional therapy plays a complementary part in the healing of various ailments [12,13]. A healthy diet is one in which macronutrients are consumed in proportions that are adequate to support energy and physiological needs, without an excessive intake, while providing sufficient micronutrients and hydration to meet the body’s physiological needs. Macronutrients (carbohydrates, proteins, and lipids) provide the energy necessary for the cellular processes required for daily functioning. Micronutrients (vitamins and minerals) are required in relatively small amounts for the growth, development, metabolism, and normal physiological functioning of the body [14,15,16].

A healthy diet (Figure 1) combined with a daily exercise program and a sufficient amount of sleep, helps to maintain the body weight within normal limits, as well as to reduce the risk of many chronic diseases, such as obesity, cardiovascular diseases, dental, diabetes, cancer, skeletal diseases, etc. [17,18].

It is important to pay particular attention to the sources used to obtain food raw materials, as food must be as free from contaminants or degradation products, and as minimally processed, as possible, to avoid the introduction of unwanted, potentially toxic compounds into the body. It is ideal to consume vegetables and fruits as naturally as possible and unsprayed; meat, eggs, fish, seafood, and dairy products from safe sources; and foods that are as minimally processed and free of additives as possible. This is unfortunately, nowadays, quite difficult to achieve, but not impossible. Avoiding the contamination of food products by various toxic substances from the environment (pesticides, heavy metals, microplastics, etc.) represents an essential element for increasing consumer safety [19,20,21].

Among the unhealthy foods are junk food products, which are hyper-caloric food products, poor in valuable nutrients for the body, and heavy in hydrogenated fats, which are rich in trans acids [22,23,24]. The long-term excessive consumption of these foods, correlated with a low consumption of fiber and antioxidants provided mainly by vegetable products, and with an unhealthy and predominantly sedentary lifestyle is one of the main causes of alterations in the quality of life, and the appearance of the metabolic syndrome with complications from an early age (Figure 2). Industrially ultra-processed foods, rich in various synthetic additives, are also among those that lead to excessive consumption, causing the appearance of insulin resistance, obesity, allergic phenomena, food intolerances, decreased immunity, and intestinal inflammatory syndrome [25,26,27,28].

The term junk food usually refers to foods with little nutritional value, high in calories, fat, sugar, salt, or caffeine, and can include candy, chips, cookies, French fries, chewing gum, hamburgers, hot dogs, shawarma, ice cream, juice (especially sweetened carbonated beverages), and many packaged sweets [29]. These food products are found especially within the category of ultra-processed products, rich in food additives.

Ultra-processed foods can contribute to weight gain through their nutritional profile (a low intake of valuable nutrients, but rich in caloric substances), and by replacing low-calorie, nutritious, and minimally processed foods in the diet. Intense flavors, resulting from high levels of fat, salt, sugar, and artificial flavorings make ultra-processed foods highly palatable, potentially altering endogenous satiety mechanisms. Ultra-processed foods are, on average, higher in calories and less filling than minimally processed foods. Because satiety mechanisms are more sensitive to volume than caloric content, foods with a higher energy density may facilitate an excessive energy intake [30,31].

The higher energy density and oro-sensory characteristics of ultra-processed foods (softer, less fibrous, easy-to-chew textures) may also allow for greater energy intake in a shorter period of time. Experimental studies suggest that increased feeding rates may lead to an increased energy intake, potentially due to a delayed signaling of satiety [32].

Considering that junk food products are products with a low nutritional value, and are hypercaloric, with the risk of causing a food addiction to develop and significantly alter the state of health, especially when they predominate in the diet, and are accompanied by an unhealthy lifestyle, it is very important to evaluate some aspects related to their inclusion in the population’s diet. By disseminating a questionnaire among the population aged 18 and over in Romania (adult population), a cross-sectional observational study was carried out that followed the consumption of junk food, along with an evaluation of other health risk factors represented by eating habits and lifestyle.

## 2. Materials and Methods

### 2.1. Study Design

With the help of a questionnaire with 49 questions, structured in such a way as to collect information related to the socio-demographic and anthropometric characteristics of the participants, along with the type and frequency of consumption of various junk food products, but also information related to diet and lifestyle characteristics, a cross-sectional observational study was carried out. The Google Forms web platform was used to disseminate the questionnaire in the online environment, through social networks, WhatsApp, and personal and institutional email addresses (higher education institutions and large companies that agreed to participate in the study). The questionnaire was addressed to people aged 18 and over, participation was voluntary; with the assurance of all conditions of confidentiality regarding personal data and identity, each respondent agreed to participate in the study without any coercion (informed consent), and without any discrimination related to sex, religion, or political beliefs, or of any other nature.

### 2.2. Questionnaire Validation

For the validation of the questionnaire, seven experts collaborated, and initially tested the questionnaire, in the pilot phase, on 200 volunteers. At this stage, the consistency, accuracy, and clarity of the content of the questions were improved, and ambiguous parts of the questionnaire were removed and modified in such a way that the information processed based on the collected data provided the most complete picture possible of the study carried out. The Cronbach’s alpha (α) coefficient determined was 0.83, a value that reflects a good internal consistency for the work tool used in the study [33,34,35,36].

### 2.3. Statistical Analysis

The Kolmogorov–Smirnov test was used for the normality checking of the continuous variables. The categorical variables are presented with frequencies (percentages). Differences in the categorical variables were analyzed using a Chi-square test. If the overall Chi-square test resulted in a *p*-value < 0.05, then post hoc comparisons were performed using a Bonferroni test (following the chi-square tests) [37,38,39].

Correspondence analysis (CA) was performed to visualize the relationships between age and BMI and participant responses. Furthermore, a multivariate multinomial logistic regression, with odds ratio, was performed to assess the association between junk food consumption and adherence to healthy diet categories and socio-demographic, anthropometric characteristics (age group, gender, BMI group, residence area, education level, and employment status).

Finally, a multivariable multinomial logistic regression was employed to assess the association between adherence to healthy diet and lifestyle (sleeping habits, smoking, physical activity, frequency of fast food consumption). Statistical analysis was performed using Statistical Package Software for Social Science, version 23 (SPSS Inc., Chicago, IL, USA) and XLSTAT (version 2020, Addinsoft, New York, NY, USA). *p*-values of less than 0.05 were considered statistically significant.

## 3. Results

### 3.1. Socio-Demographic and Anthropometric Data

The study carried out, based on the dissemination of the questionnaire, registered a total number of 1719 valid answers that came from 81.4% female respondents and 18.6% male respondents. The anthropometric data (weight and height) were used to calculate BMI using the Quetelet equation [body mass (kg)/height (m^2^)], and interpreted according to the criteria of the World Health Organization [40,41,42]. After processing the anthropometric data, we observed that most of the respondents are of normal weight (1098), among women in particular (906). The socio-demographic and anthropometric characteristics of the respondents are presented in Table 1.

Most of the respondents are between 18 and 23 years old (48.1%), and 62% of them are students. Most respondents live in an urban area (80.9%) and are people with a secondary education (baccalaureate diploma) (41.4%). In addition, from the processing of the anthropometric data (Figure 3a), overweight (39.4%) and obese (19.4%) people are found among the respondents aged over 45 in the largest number, compared to those aged between 18 and 23 years (13.4% overweight and 5.6% obese) or aged between 24 and 35 years (15.6% overweight, 6% obese). Respondents with a normal weight are the majority in the age groups 18–23 (68.2%) and 24–35 (68.7%).

Higher percentages of overweight (29.2%) and obese (16.9%) respondents are found among the male respondents (Figure 3b). Underweight people are more common among the female respondents (11.8%) compared to the male participants (2.4%).

### 3.2. Consumption of Junk Food Products

The evaluation of the consumption of junk food products was based on four categories of food products: sweets (cakes, candies, ice cream, packaged sweet products, chewing gum), drinks (sweetened carbohydrates, sweetened non-carbohydrates, energy drinks, coffee), fast foods (hotdogs, hamburgers, fries, wrapped sandwiches, shawarma) and salty snacks (snacks, pastries, crisps). Each positive response related to consumption was quantified with 1, while each negative response related to the consumption of a certain indicated junk food product was quantified with 0. A number of 162 (9.4%) respondents said that they do not consume any of the junk food products indicated.

Chi-square analysis and correspondence analysis (CA) were conducted for the anthropometric variables of BMI and age.

For the age groups, the bi-plot indicates that 97.01% of the variability observed can be attributed to the two main components for F1 (88.67%) and F2 (8.34%). CA shows a significant difference (χ^2^ = 68.66 and *p* < 0.001) between age groups, and the 17 types of junk food products (Figure 4). The 24–35 and 36–45 age groups show similar profiles in terms of junk food consumption.

Overall, the most consumed junk food products are fried potatoes (46.2%) and pastries (41.4%). The comparisons of the column proportions indicate that the percentage of respondents who consume junk food products, younger than 23 years old, was significantly higher than those older than 35 years old (Table 2). Only 73 of the respondents declare that they consume hot dogs as a junk food product, with 64.4% being in the 18–23 age group. Energy drinks are also consumed mainly by young respondents (76.6%).

For BMI, the biplot indicated that 95.67% of the variability observed could be attributed to the two principal components for F1 (85.28%) and F2 (10.39%). The CA showed a significant difference (χ^2^ = 72.33 and *p* < 0.001) between BMI and the 17 types of junk food products (Figure 5). Obese and overweight respondents present similar profiles related to the consumption of junk food products.

The comparisons of the column proportions indicated that underweight people prefer the consumption of salty products (chips 39.9%, snacks 36.9%, and pastries 57.1%) in a significantly higher percentage compared to the other categories (Table 3).

To determine the level of junk food consumption, all the food products associated with the four categories (sweets, sweetened drinks, fast food products, and salty snacks) were summed, to obtain a total number of 17 products, with this being considered the maximum value. The classification was made as follows: a total score less than or equal to 4 was classified as low consumption, the score between 5 and 9 was classified as medium consumption, and the score greater than 9 was considered high consumption of junk food (Table 3).

Multiple linear regression analysis was used to investigate the association between fast-food consumption and the sociodemographic and anthropometric variables. In the multiple logistic regression analysis, three variables were significant to predict for a higher junk food consumption (Table 4): age: χ^2^ = 18.082, *p* = 0.032, aender: χ^2^ = 9.897, *p* = 0.019, BMI: χ^2^ = 17.127, *p* < 0.047, and level of education: χ^2^ = 76.205, *p* < 0.001.

Female respondents show a lower tendency toward an increased consumption of junk food products (OR = 0.703, 95%(CI): 0.19–0.95, *p* = 0.011) compared to male respondents. Participants between the ages of 18 and 23, with secondary education with/without a high school diploma, and those who are obese, show an approximately three-times-higher tendency to consume junk food products.

### 3.3. Adherence to Healthy Diet and Lifestyle Correlated with Junk Food Consumption

The data were completed with a new categorical variable named “Adherence to a healthy diet”, which was constructed based on the answers to questions that tracked the frequency of healthy food consumption (vegetables and fruits, eggs, dairy products, fish, seafood, meat, virgin and extra-virgin vegetable oils, whole grains, moderate consumption of sweets, alcohol, proper hydration), with all questions having answers from 1 (very little or not at all, regarding the frequency of consumption) up to 5 (a lot or always, frequency of consumption). Summing the answers formed a raw score, which we then scaled into a T score (standardized), with an average of 53.48 and standard deviation of 7. Scores above 60 indicated maximum values for adherence to a healthy diet, and those below 50 indicated an adherence to an unhealthy diet (Table 5).

A multiple linear regression analysis was used to investigate the association between the adherence to heathy diet (dependent variable) and the socio-demographic and anthropometric variables. In multiple logistic regression analysis, four variables were significant to predict for a higher junk food consumption (Table 6). These were for age: χ^2^ = 39.931, *p* < 0.01; for level of education: χ^2^ = 27.084, *p* = 0.001 and for BMI: χ^2^ = 15.566, *p* = 0.016.

Participants aged between 18 and 23 years have the highest tendency to adhere to an unhealthy diet (OR = 5.35, 95%(CI): 1.89–9.12, *p* = 0.002), or a moderately healthy diet (OR = 2.46, 95% (CI): 1.22–4.96, *p* = 0.011), compared to those aged over 45 years.

Participants with a secondary education (baccalaureate degree) have an increased tendency toward an unhealthy diet compared to participants with a higher education (OR = 1.612 95%(CI):1.13–2.29, *p* = 0.008). Additionally, obese participants have an increased risk of having an unhealthy diet (OR = 1.49 95%(CI): 1.001–2.99, *p* = 0.049) compared to normal weight respondents.

Regarding lifestyle habits, we studied physical activity, smoking, sleeping habits, and the frequency of junk food consumption (Table 7).

The z-test indicated that a greater proportion of participants with an unhealthy diet do not exercise (31.5%), smoke excessively (30.59%), sleep less than 7 h per night (42.92%), and frequently consume junk food products (38.81%) than those who have a predominantly healthy diet (Table 7 and Table 8).

Multivariate logistic regression with healthy diet as the reference category was performed (Table 8).

Respondents who consume junk food products daily have an increased risk of adhering to an unhealthy (OR = 8.31, 95% (CI): 3.55–13.86, *p* < 0.001) or moderately healthy (OR = 7.73, 95% (CI)) diet: 2.53–12.49, *p* = 0.002) compared to those who declared that they did not consume junk food products.

Those who declare that they smoke excessively daily have a 4.65 (95% (CI): 3.06–7.07, *p* < 0.001)-times-higher risk of adhering to an unhealthy diet, and a 1.89 (95% (CI): 1.27–2.81, *p* = 0.002) times higher to adhere to a moderately healthy diet, compared to non-smokers.

Respondents who exercise daily for at least one hour have a lower risk of adhering to an unhealthy diet (OR = 0.06, 95% (CI): 0.3–0.12, *p* < 0.001) or a moderately healthy diet (OR = 0.22, 95% (CI): 0.3–0.38, *p* < 0.001) compared to those who do not do sports. Moreover, even those who do sports very rarely have a lower risk of adhering to an unhealthy diet (OR = 0.42, 95% (I CI): 0.26–0.68, <0.001).

Study participants who declared that they have insomnia have a higher risk of adhering to an unhealthy diet, compared to those who sleep 7–8 h per night (OR = 2.41, 95% (CI): 1.35–4.3, *p* = 0.003).

The main foods that predominate in the daily diet according to the answers recorded are meat (30.5%) and vegetables and fruits (28.3%). The least consumed are fish and seafood dishes (1%). A significantly higher percentage of respondents who have a healthy diet predominantly consume vegetables and fruits (54.8%) (Figure 6).

The most consumed liquid is still mineral water (46.9%), followed by drinking water (24.1%). Only 0.9% of respondents consume alcoholic beverages often. About 74% of respondents with a healthy diet consume mostly plain water (Figure 7).

A significant statistical association was found between the frequency of fast food product consumption and anthropometric data (age and BMI). More than 60% of the respondents in the age category 18–23 have a regular consumption of junk food products from daily to at least once a week (Figure 8a), and those between 24 and 35 years, about 30%. The highest percentage of people who consume these products very rarely or not at all is among respondents over 45 years of age.

Obese and overweight people who consume junk food products regularly (daily or at least once a week) amount to a percentage between 22.43% and 31% (Figure 8b).

According to the centralized data from the respondents, the main reasons underlying the choice of consuming fast food or ready-to-eat products are (Figure 9) a lack of time (50.4% of respondents), the pleasure of consuming such products (38.9%), satisfying the need for sweets (35.4%), and convenience (32.3%).

Most of the respondents declared that they eat 1–2 meals a day without a fixed schedule (Figure 10), with the highest percentage being recorded among obese (57.93%) and underweight (52.98%) people.

The majority of underweight respondents (Figure 11) believe that they eat chaotically and insufficiently (51.9%), and the majority of obese respondents believe that they eat chaotically, without measure (59.31%).

Regarding the way of serving meals, 44.14% of obese respondents and 37.66% of overweight respondents declared that they eat in a hurry, while 32.41% of obese people and 27.92% of overweight people do something else during the meal (Figure 12).

According to the answers received from the respondents, the junk food products (Figure 13) that are most indicated to cause a consumption addiction, apart from coffee, (34.8%) are fried potatoes, pastries, sweets, and sweetened drinks.

Most of the respondents declared that they take part in sports very rarely or not at all; the greatest tendency toward sedentariness is shown by obese and overweight people (Figure 14a). Regarding the duration of sleep, the highest percentage of respondents who rest insufficiently is found among obese people (Figure 14b).

The most frequent problems (Figure 15) that alter the respondents’ quality of life, according to the centralized data, are fatigue (62.4% of the respondents), frequent nervousness (37.5%), agitation (32.9%), overeating emotional (29.30%), and depressive (27.6%) and anxiety states (21.9%).

## 4. Discussion

Following the dissemination of the questionnaire, 1719 valid answers were collected; most of the respondents were between 18 and 23 years old (48.1%), and 29.1% of the respondents were between 24 and 35 years old (Table 1). It is also noted that the majority of study participants are female (81.4%), have secondary education (41.4%), come from an urban environment (80.9%), and are pupils or students (62%). Regarding the anthropometric characteristics, 63.8% of the respondents are a normal weight, 18% overweight, 8.4% obese, and 9.8% underweight.

During the study, an identification of the most-consumed food products of the junk-food type was conducted (Figure 4), and it was found that the top preferences are fried potatoes (46.2%), pastry products (41.4%), chewing gum (33.9%), and sweetened carbonated drinks (32.9%), along with coffee (62.8%). Regarding the frequency of consumption of junk food products (Figure 4, Table 2), approximately 51% of the respondents declared that they consume them daily or weekly, with the majority being from the category of the very young (18–23 years old). There are numerous studies in the specialized literature that draw attention to the increased consumption of junk food products, especially among teenagers and young people, but also to the negative consequences on the body in the long term: the increase in the rate of obesity, and the incidence of inflammatory bowel diseases, allergies and food intolerances, and type 2 diabetes. The most incriminated are sweet products, sweetened drinks, pastry products, and snacks [43,44,45].

Although 93.9% of respondents declared that they do not consider fast food or packaged ready-to-eat food products as healthy products, the main reasons for consuming them are a lack of time, convenience, and food addiction (Figure 9). In recent times, the alarm bells raised by specialists have intensified regarding the danger of food addiction, especially that caused by ultra-processed foods, on the health of consumers, with serious consequences, especially due to the adoption of a lifestyle with reduced physical activity. These food addictions have generated an explosion in the obesity rate and the increase in the incidence of metabolic diseases, especially from an early age [46,47]. Junk food products are rich in additives, some of which (such as food sweeteners, flavorings, monosodium glutamate, etc.) play an important role in the development of food addictions [48,49,50].

The analysis of junk food consumption was made in correlation with adherence to a healthy diet (which implies an optimal consumption of vegetables and fruits, whole grains, nutritious foods such as fish, seafood, dairy products, and eggs, and predominantly unsaturated fats, as well as the optimal hydration of the body), but also with adherence to a healthy lifestyle (which implies adequate rest, an active life in terms of physical activities, and avoiding the consumption of alcohol and tobacco, and other unhealthy habits). Based on the quantification of the answers that followed the evaluation of the two components, it was found that 18.67% of the respondents (Table 5 and Table 6) have an increased adherence to healthy eating (more women, the elderly compared to the young, and those of a normal weight, in particular), while 25.48% have an increased adherence to an unhealthy diet (especially people aged up to 24). People with low adherence to a healthy lifestyle (Table 7 and Table 8) also show an increased adherence to an unhealthy lifestyle (reduced physical activity, inadequate rest, and increased consumption of alcohol or cigarettes).

From the analysis of the centralized data, it was found that 62.9% of the respondents declared that they consume one portion of vegetables (100 g) per day or rarely, and 64.5% declared that they consume only one portion of fruit per day or rarely, which is quite worrying. Vegetables and fruits offer an important supply of antioxidants, vitamins, minerals, and fiber, which is particularly important for the health of consumers. Additionally, a very low consumption of fish and seafood was noted among the respondents; 33.3% declared that they consumed it very rarely, and 38.7% only 2–3 times a month. With regard to the degree of hydration of the body, there is an insufficient intake of water; 12.3% consume less than one liter of water per day, and 33.7% approximately one liter. On the other hand, 43.9% of the respondents consume sweetened drinks daily or weekly.

The trend of a reduced consumption of vegetables and fruits among young people is signaled by numerous scientific studies [51,52,53] that also highlight the negative consequences on their health (an increase in the incidence of cardiovascular diseases, diabetes, obesity; at the same time, there is a worldwide concern in numerous states toward stimulating the consumption of vegetables and fruits among young people, precisely to ensure an optimal intake of valuable nutrients [54,55,56].

There is also an increased trend toward sedentarism; 58.4% of the respondents declared that they do sports very rarely or not at all. Regarding cigarette consumption, 27.7% of respondents smoke heavily, and regarding rest, 39.1% of respondents do not rest enough.

Regarding the state of the immune system, 22.2% of the respondents declared that they have a weakened or unbalanced immune system, and 23% of the study participants frequently use various methods to strengthen their immune system. Regarding the need for the services of a nutritionist, 29.1% of the respondents stated that they need a specialist to help them choose healthy foods, while 19.7% of the respondents need the help of a nutritionist to eat balanced meals. Another worrisome aspect to point out: it was found that 43.4% of the respondents do not usually evaluate their health status, or do it very rarely.

Unfortunately, the consumption of junk food products is usually accompanied by the consumption of sweetened beverages, and even smoking or alcoholic beverages (which adds a series of dangerous compounds to the body, and a significant increase in calories), and less vegetable products which would have ensured an intake of important fibers and antioxidants for the detoxification of the body [57,58,59]. As a result, in the long term, unhealthy and unbalanced food associations are also the basis of micronutrient malnutrition, strongly altering the state of health [60].

There are a number of limitations to the study, related to the low participation of the population over 35 years old, probably because they were not very receptive to being included in the study, and because they considered that they are not very used to consuming junk food products, as well as a lower participation among the male population. The most important aspect of the study, however, is the increased participation of young people in particular. The collected data offer the possibility of evaluating some important aspects related to the eating habits and lifestyle of the young population, the active population, both physically and in terms of demographic development potential. Thus, it is found that among the young population there is a low consumption of foods with high nutritional value (vegetables, fruits, fish, seafood), an inadequate hydration of the body, a tendency toward a sedentary life, and an increased consumption of ultra-processed foods. In the long term, there is a risk of major imbalances. The signals are already there; although most of the respondents are young people, they face serious problems that alter their quality of life in a fairly large percentage: fatigue, frequent nervousness, depression, anxiety, and even problems with the immune system. In this sense, there is a need to intensify social–educational programs, to stimulate physical activity from an early age, as well as to stimulate the consumption of vegetables, fruits, fish, seafood, and proper hydration of the body.

## 5. Conclusions

The consumption of junk food becomes dangerous for health when it is accompanied by a sedentary lifestyle, and is especially associated with other hypercaloric foods, such as sweetened or alcoholic beverages. Physical activity helps to offset excess calories, and the regular consumption of vegetable and fruit products counterbalances the absence or deficiency of important nutrients for the body, such as fibers, antioxidants, vitamins, minerals, and unsaturated fats.

## Figures and Tables

**Figure 1 nutrients-15-03591-f001:**
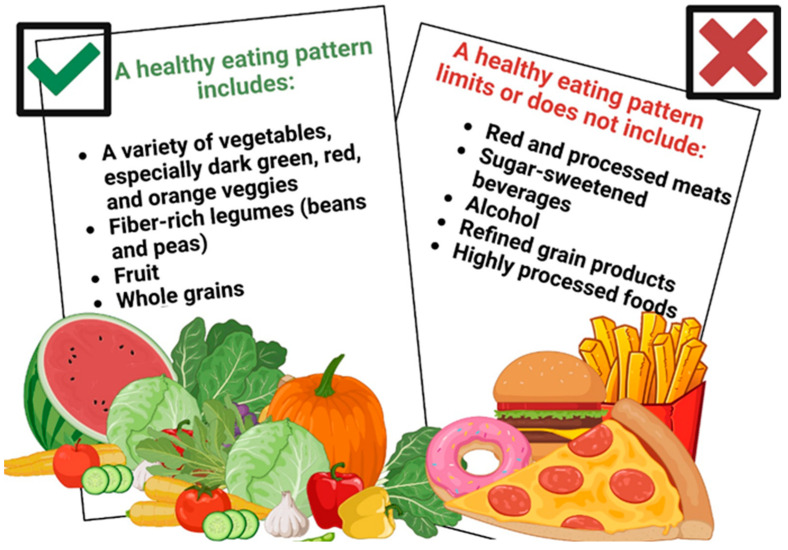
Characteristics of a healthy diet. Created using BioRender.com.

**Figure 2 nutrients-15-03591-f002:**
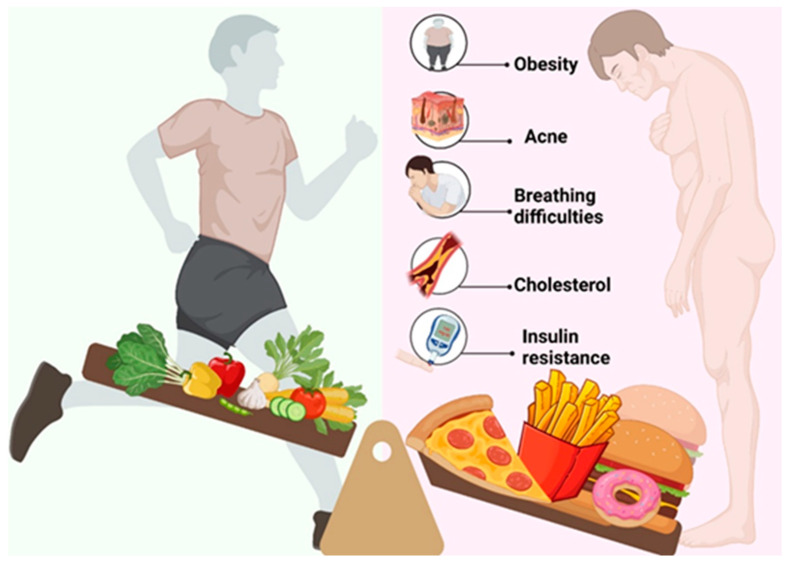
The risks of the excessive consumption of junk food products. Created using BioRender.com.

**Figure 3 nutrients-15-03591-f003:**
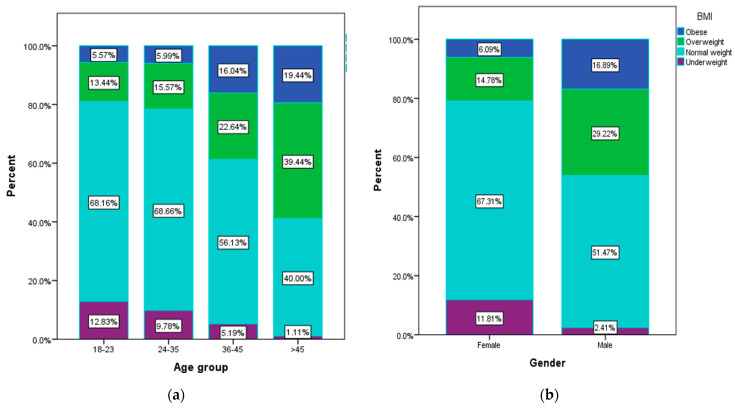
Weight status according to (**a**) age group (χ^2^ = 160.49, *p* < 0.001), and (**b**) gender (χ^2^ = 112.22, *p* < 0.001).

**Figure 4 nutrients-15-03591-f004:**
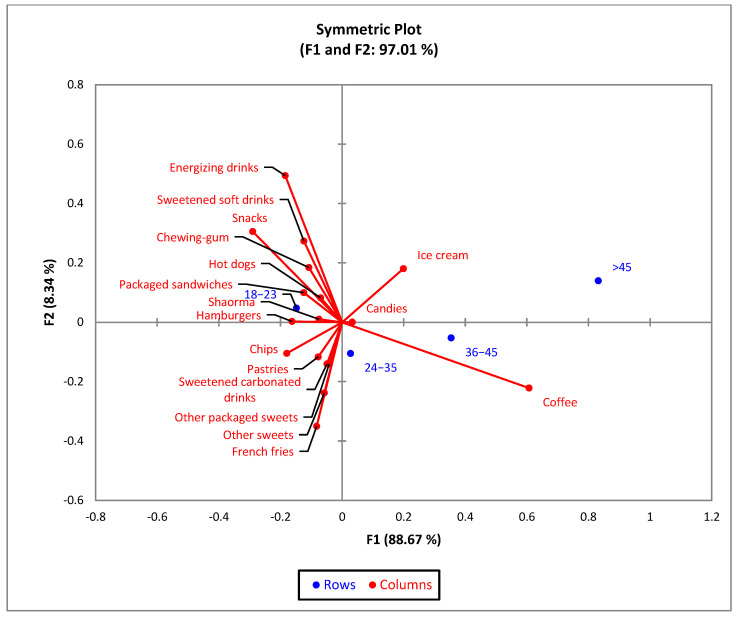
The first two dimensions of the correspondence analysis (CA) symmetric plot, using age (18–23, 24–35, 36–45, and 45 and older) as rows, and all the 17 answers as columns.

**Figure 5 nutrients-15-03591-f005:**
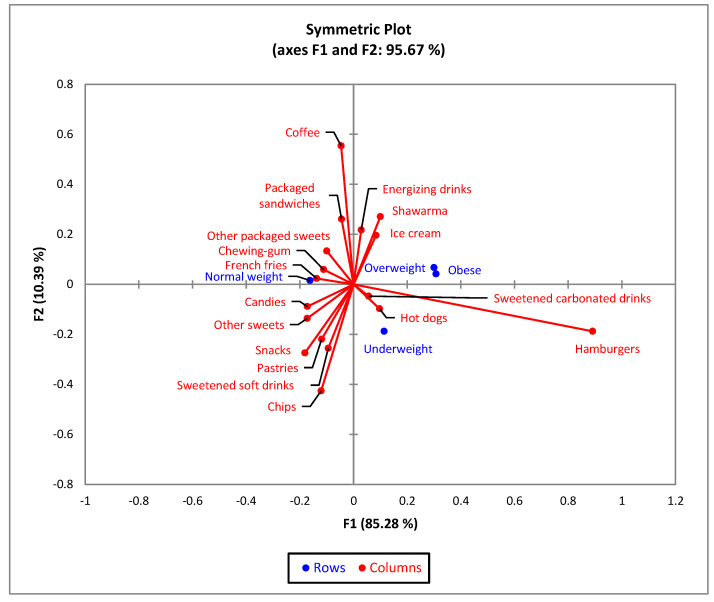
The first two dimensions of the correspondence analysis (CA) symmetric plot using BMI (Underweight, Normal weight, Overweight and Obese) as rows, and all 17 answers as columns.

**Figure 6 nutrients-15-03591-f006:**
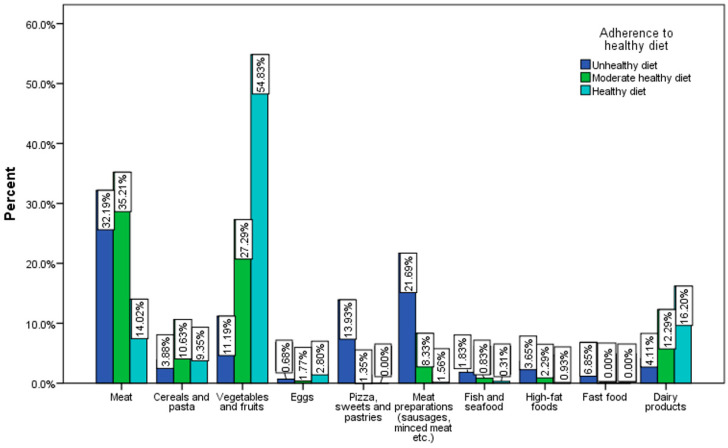
Food category predominates in the daily diet by the adherence to a healthy diet (χ^2^ = 57.78, *p* < 0.001).

**Figure 7 nutrients-15-03591-f007:**
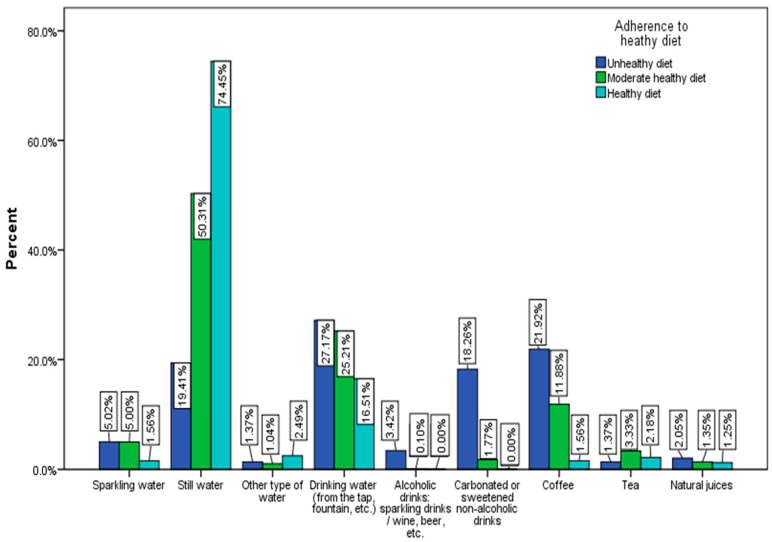
Drink category predominates in the daily diet by the adherence to a healthy diet (χ^2^ = 42.02, *p* < 0.001).

**Figure 8 nutrients-15-03591-f008:**
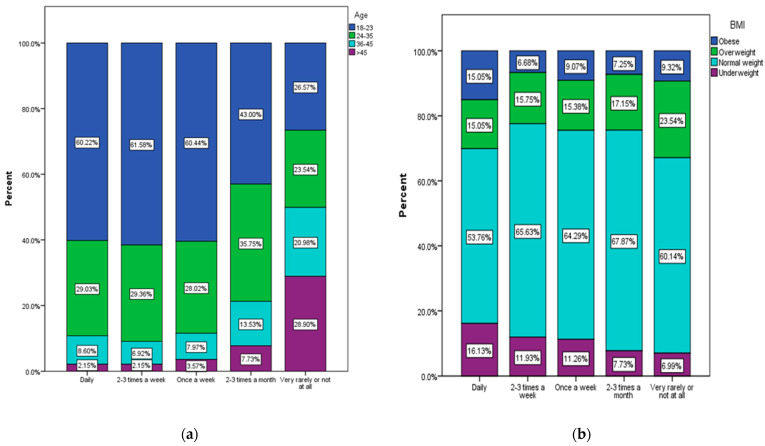
Frequency of junk food product consumption by (**a**) age (χ^2^ = 74.28, *p* < 0.001), and (**b**) BMI (χ^2^ = 33.67, *p* = 0.001).

**Figure 9 nutrients-15-03591-f009:**
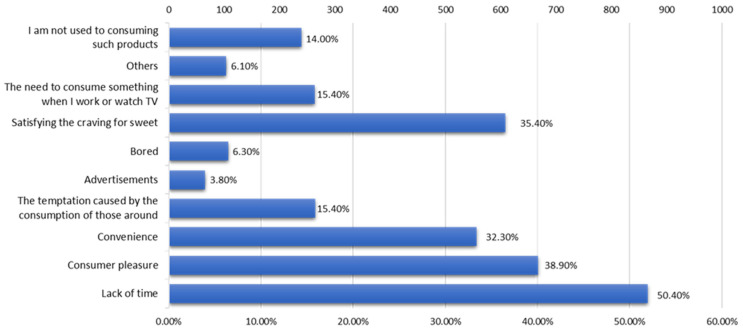
The main reasons underlying the consumption of junk food products.

**Figure 10 nutrients-15-03591-f010:**
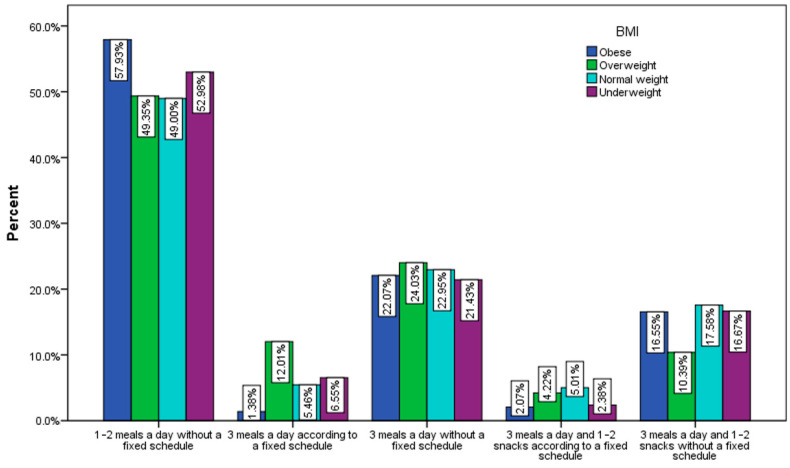
Distribution of meals per day by BMI (χ^2^ = 37.2, *p* < 0.001).

**Figure 11 nutrients-15-03591-f011:**
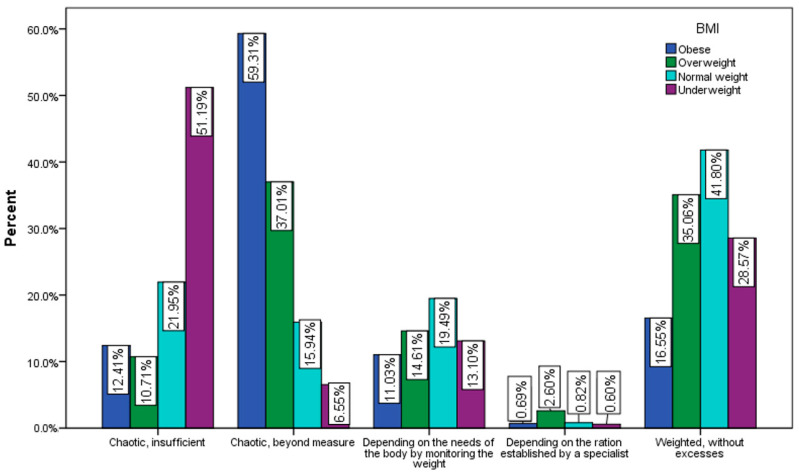
The amount of food consumed by BMI (χ^2^ = 89.1, *p* < 0.001).

**Figure 12 nutrients-15-03591-f012:**
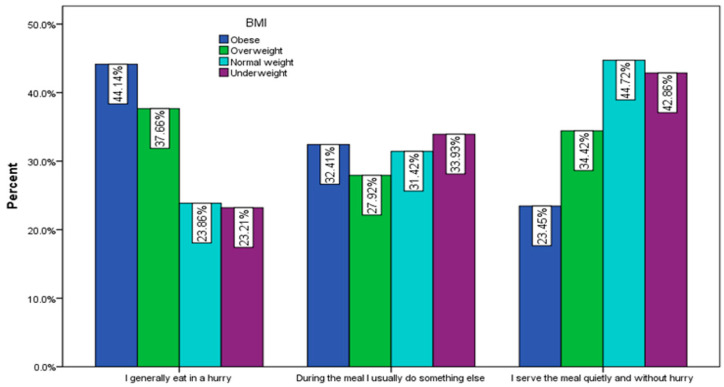
Habits during meals by BMI (χ^2^ = 51.45, *p* < 0.001).

**Figure 13 nutrients-15-03591-f013:**
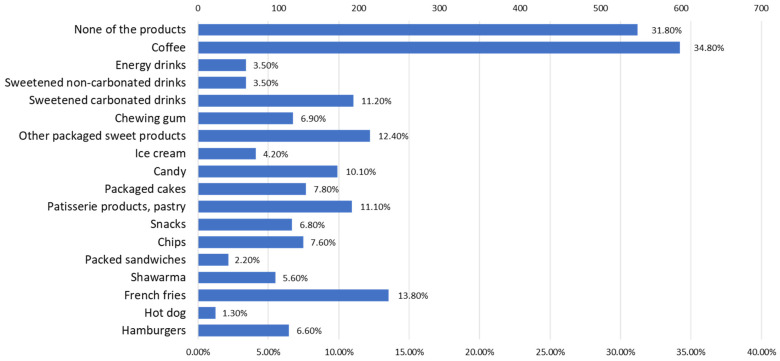
Evidence of consumption addictions for different junk food products.

**Figure 14 nutrients-15-03591-f014:**
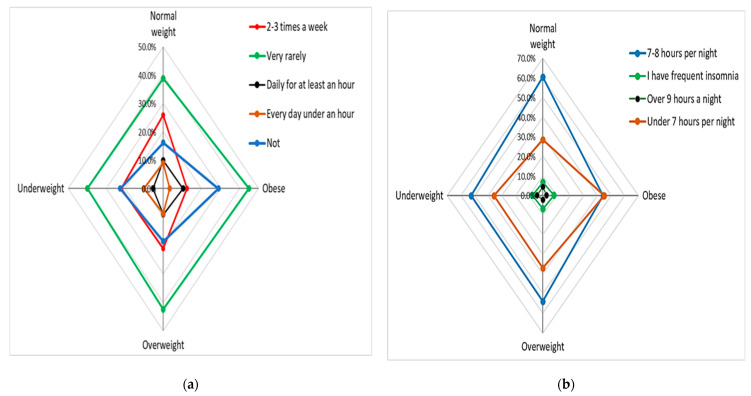
(**a**) Physical activity by BMI (χ^2^ = 34.85, *p* < 0.001); (**b**) sleep duration by BMI (χ^2^ = 26.49, *p* = 0.002).

**Figure 15 nutrients-15-03591-f015:**
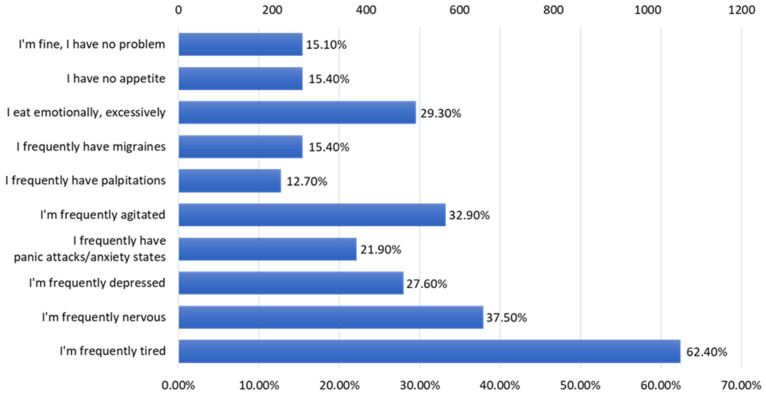
The problems that alter the quality of life.

**Table 1 nutrients-15-03591-t001:** Socio-demographic and anthropometric characteristics of the respondents (n = 1719).

	Total Population n (%)	Male n (%)	Female n (%)
	1754 (100)	327 (18.6)	1424 (81.4)
Age (years)		*p* < 0.0214	
18–23	826 (48.1)	204 (54.7)	622 (46.2)
24–35	501 (29.1)	94 (25.2)	407 (30.2)
36–45	212 (12.3)	36 (9.7)	176 (13.1)
>45	180 (10.5)	39 (10.4)	141 (10.5)
Residence area		*p* = 0.5139	
Urban area	1390 (80.9)	306 (82.04)	1084 (80.5)
Rural area	329 (19.1)	67 (17.96)	262 (19.5)
Level of education		*p* = 0.0008	
General/primary studies	89 (5.2)	32 (8.6)	57 (4.2)
Secondary education (baccalaureate degree)	712 (41.4)	172 (46.1)	540 (40.1)
Post-secondary studies	112 (6.5)	21 (5.6)	91 (6.8)
Higher education (bachelor’s degree)	482 (28)	88 (23.6)	394 (29.3)
Postgraduate studies (master’s degree, residency, doctorate, other specializations)	324 (18.9)	60 (16.1)	264 (19.6)
Employment status		*p* = 0.0003	
Unemployed	19 (1.1)	9 (2.4)	10 (0.7)
Socially assisted	6 (0.3)	4 (1.1)	2 (0.1)
Householder	30 (1.7)	3 (0.8)	27 (2)
Retired	16 (0.9)	7 (1.9)	9 (0.7)
Student	1066 (62)	242 (64.9)	824 (61.2)
Teleworking	27 (1.6)	5 (1.3)	22 (1.6)
I go to work every day	488 (28.4)	86 (23.1)	402 (29.9)
I work in a mixed regime (telework and commuting)	67 (3.9)	17 (4.6)	50 (3.7)
Body mass index (BMI)		*p* < 0.0001	
Within normal limits (18.5–24.9)	1098 (63.8)	192 (51.5)	906 (67.3)
Overweight category (25–29.9)	308 (18)	109 (29.2)	199 (14.8)
Underweight category (<18.5)	168 (9.8)	9 (2.41)	159 (11.8)
Obese (≥30)	145 (8.4)	63 (16.9)	82 (6.1)

**Table 2 nutrients-15-03591-t002:** Type of consumed junk food products according to age, and test of equality for column proportions (z-test).

Type of Consumed Junk Food Products	Age							
	18–23 (a)		24–35 (b)		36–45 (c)		>45 (d)	
	n	%	n	%	n	%	n	%
Hamburgers	223 ^b,c,d^	63.4	99 ^c,d^	28.1	23	6.5	7	2.0
Hot dogs	47 ^c^	64.4	21	28.8	2	2.7	3	4.1
French fries	454 ^c,d^	56.3	252 ^c,d^	31.3	74 ^d^	9.2	26	3.2
Shawarma	149 ^c,d^	60.6	70 ^d^	28.5	18	7.3	9	3.7
Packaged sandwiches	144 ^b,c,d^	64.9	57 ^d^	25.7	16	7.2	5	2.3
Chips	261 ^b,c,d^	62.3	122 ^c,d^	29.1	30 ^d^	7.2	6	1.4
Snacks	313 ^b,c,d^	70.3	109 ^c,d^	24.5	18	4.0	5	1.1
Pastries	413 ^b,c,d^	58.0	213 ^c,d^	29.9	57	8.0	29	4.1
Other packaged sweets	147 ^c,d^	56.5	83 ^c,d^	31.9	19	7.3	11	4.2
Candies	231 ^c,d^	55.5	124 ^d^	29.8	34	8.2	27	6.5
Ice cream	125 ^c,d^	50.4	66 ^c,d^	26.6	29	11.7	28	11.3
Other sweets	267 ^c,d^	55.9	159 ^c,d^	33.3	31	6.5	21	4.4
Chewing-gum	360 ^b,c,d^	61.9	158 ^c,d^	27.1	38	6.5	26	4.5
Sweetened carbonated drinks	325 ^b,c,d^	57.5	161 ^d^	28.5	59 ^d^	10.4	20	3.5
Sweetened soft drinks	201 ^b,c,d^	65.3	81 ^c,d^	26.3	12	3.9	14	4.5
Energizing drinks	131 ^b,c,d^	76.6	28	16.4	9	5.3	3	1.8
Coffee	450	41.7	349 ^a^	32.3	144 ^a^	13.3	136 ^a^	12.6
Do not consume	52	32.1	35	21.6	33 ^a,b^	20.4	42 ^a,b^	25.9

Values in the same row not sharing the same superscript were significantly different at *p* < 0.05 in the two-sided test of equality for column proportions (z-test). Tests were adjusted for all pairwise comparisons within a row of each innermost sub-table using the Bonferroni correction.

**Table 3 nutrients-15-03591-t003:** Type of consumed junk food products by age and test of equality for column proportions (z-test).

Type of Consumed Junk Food Products	BMI							
	Underweight (a)		Normal Weight (b)		Overweight (c)		Obese (d)	
	n	%	n	%	n	%	n	%
Hamburgers	44	26.2	211	19.2	60	19.5	37	25.5
Hot dogs	12	7.1	36	3.3	15	4.9	10	6.9
French fries	94 ^c^	56.0	519	47.3	127	41.2	66	45.5
Shawarma	24	14.3	138	12.6	54	17.5	30 ^b^	20.7
Packaged sandwiches	18	10.7	143	13.00	47	15.3	14	9.7
Chips	67 ^b,c,d^	39.9	266	24.2	56	18.2	30	20.7
Snacks	62 ^b,c,d^	36.9	296	27.0	61	19.8	26	17.9
Pastries	96 ^b,c,d^	57.1	451	41.1	114	37.0	51	35.2
Other packaged sweets	26	15.5	172	15.7	40	13.0	22	15.2
Candies	51 ^c^	30.4	279	25.4	57	18.5	29	20.0
Ice cream	27	16.1	140	12.8	46	14.9	35 ^b^	24.1
Other sweets	60 ^c,d^	35.7	318	29.0	73	23.7	27	18.6
Chewing gum	65	38.7	375	34.2	103	33.4	39	26.9
Sweetened carbonated drinks	75 ^b,c^	44.6	327	29.8	101	32.8	62	42.8
Sweetened soft drinks	45 ^b,c,d^	26.8	196	17.9	48	15.6	19	13.1
Energizing drinks	16	9.5	102	9.3	34	11.0	19	13.1
Coffee	94	56.0	678	61.7	210 ^a^	68.2	97	66.9
Do not consume	15	8.9	103	9.4	25	8.1	19	13.1

Values in the same row not sharing the same superscript were significantly different at *p* < 0.05 in the two-sided test of equality for column proportions (z-test). Tests were adjusted for all pairwise comparisons within a row of each innermost sub-table using the Bonferroni correction.

**Table 4 nutrients-15-03591-t004:** Results of the multinomial logistic regression for the junk food consumption level.

Independent Variables	Junk Food Consumption Level								
	Low			Medium			High		
	OR	95% CI	*p*	OR	95% CI	*p*	OR	95% CI	*p*
Gender									
Male	1			1			1		
Female	1.771	(1.155–2.715)	0.009	1.420	(0.909–2.217)	0.123	0.703	(0.195–0.959)	0.011
Age (years)									
18–23	0.997	(0.481–2.068)	0.994	1.474	(0.670–3.243)	0.335	2.907	(1.225–6.898)	0.015
24–35	1.312	(0.674–2.556)	0.048	2.082	(1.007–4.305)	0.048	2.670	(0.965–7.382)	0.058
36–45	0.882	(0.444–1.749)	0.719	1.234	(0.579–2.629)	0.586	0.517	(0.142–1.882)	0.317
>45	1			1			1		
Residence area									
Urban area	1			1			1		
Rural area	0.787	(0.497–1.245)	0.307	0.913	(0.568–1.468)	0.706	0.898	(0.490–1.645)	0.728
Level of education									
General/primary studies	0.847	(0.312–2.297)	0.745	1.083	(0.718–5.472)	0.186	3.304	(1.003–8.556)	0.049
Secondary education (baccalaureate degree)	1.107	(0.684–1.793)	0.678	2.607	(1.574–4.308)	<0.001	3.258	(1.693–6.273)	<0.001
Post-secondary studies	0.797	(0.378–1.683)	0.552	1.024	(0.460–2.283)	0.953	1.157	(0.387–3.46)	0.795
Higher education (bachelor’s degree)	1			1		1			
Postgraduate studies (master’s degree, residency, doctorate, other specializations)	0.655	(0.4–1.074)	0.093	0.784	(0.457–1.343)	0.375	0.897	(0.419–1.921)	0.897
Body mass index (BMI)									
Underweight category (<18.5)	0.922	(0.444–1.913)	0.922	0.643	(0.306–1.350)	0.243	0.586	(0.246–1.393)	0.226
Normal limits (18.5–24.9)	1			1			1		
Overweight category (25–29.9)	0.635	(0.285–1.411)	0.265	0.508	(0.225–1.148)	0.104	0.473	(0.179–1.249)	0.131
Obese (≥30)	0.731	(0.291–1.834)	0.504	0.617	(0.241–1.580)	0.314	3.969	(1.644–9.589)	0.025

Dependent variable: do not consume frequently junk food as reference category.

**Table 5 nutrients-15-03591-t005:** Adherence to healthy diet by age group, gender, BMI group, residence area, education level, and employment status (n = 1719).

Variable	Adherence to Healthy Diet Mean = 49.47, SD = 6.27, Min = 26, Max = 68					
	Unhealthy Diet		Medium Healthy Diet		Healthy Diet	
	n	%	n	%	n	%
Total	438	25.48	960	55.85	321	18.67
Gender (χ^2^ = 4.92, *p* = 0.085)						
Female	327	74.66	760	79.17	259	80.69
Male	111	25.34	200	20.83	62	19.31
Age (years) (χ^2^ = 50.2, *p* < 0.001)						
18–23	249	56.85	459	47.81	118	36.76
24–35	109	24.89	299	31.15	93	28.97
36–45	54	12.33	104	10.83	54	16.82
>45	26	5.94	98	10.21	56	17.45
Residence area (χ^2^ = 1.5, *p* = 0.471)						
Urban area	346	79.00	785	81.77	259	80.69
Rural area	92	21.00	175	18.23	62	19.31
Level of education (χ^2^ = 26.08, *p* = 0.001)						
General/primary studies	29	6.62	44	4.58	16	4.98
Secondary education (baccalaureate degree)	198	45.21	410	42.71	104	32.40
Post-secondary studies	30	6.85	60	6.25	22	6.85
Higher education (bachelor’s degree)	124	28.31	253	26.35	105	32.71
Postgraduate studies (master’s degree, residency, doctorate, other specializations)	57	13.01	193	20.10	74	23.05
Employment status (χ^2^ = 54.93, *p* < 0.001)						
Unemployed	8	1.83	6	0.63	5	1.56
Socially assisted	4	0.91	0	0.00	2	0.62
Householder	7	1.60	16	1.67	7	2.18
Retired	1	.23	7	0.73	8	2.49
Student	299	68.26	600	62.50	167	52.02
Teleworking	7	1.60	11	1.15	9	2.80
I go to work every day	106	24.20	281	29.27	101	31.46
I work in a mixed regime (telework and commuting)	6	1.37	39	4.06	22	6.85
Body mass index (BMI) (χ^2^ = 16.39, *p* = 0.012)						
Underweight	48	10.96	94	9.79	26	8.10
Normal weight	266	60.73	619	64.48	213	66.36
Overweight	69	15.75	177	18.44	62	19.31
Obese	55	12.56	70	7.29	20	6.23

**Table 6 nutrients-15-03591-t006:** Results of the multinomial logistic regression for the adherence to healthy diet.

Independent Variables	Unhealthy Diet			Medium Healthy Diet		
	OR	95% CI	*p*	OR	95% CI	*p*
Gender						
Male	1			1		
Female	0.705	(0.496–1.002)	0.051	0.910	(0.662–1.250)	0.559
Age (years)						
18–23	5.352	(1.893–9.128)	0.002	2.468	(1.227–4.964)	0.011
24–35	3.045	(1.049–5.836)	0.040	1.908	(0.926–3.931)	0.080
35–45	1.786	(0.601–3.305)	0.297	1.208	(0.578–2.525)	0.616
>45	1			1		
Residence area						
Urban area	1			1		
Rural area	1.111	(0.775–1.592)	0.567	0.665	(0.93–1.675)	0.665
Level of education						
General/primary studies	1.535	(0.791–2.980)	0.206	1.141	(0.617–2.113)	0.674
Secondary education (baccalaureate degree)	1.612	(1.134–2.292)	0.008	1.636	(1.196–2.238)	0.002
Post-secondary studies	1.155	(0.628–2.122)	0.643	1.132	(0.660–1.940)	0.652
Higher education (bachelor’s degree)	1			1		
Postgraduate studies (master’s degree, residency, doctorate, other specializations)	0.652	(0.423–1.005)	0.053	1.082	(0.762–1.538)	0.659
Body mass index (BMI)						
Underweight category (<18.5)	0.676	(0.406–1.127)	0.133	0.804	(0.507–1.275)	0.354
Normal limits (18.5–24.9)	1			1		
Overweight category (25–29.9)	0.603	(0.335–1.085)	0.091	0.790	(0.469–1.1331)	0.375
Obese (≥30)	1.490	(1.001–2.999)	0.049	0.968	(0.500–1.873)	0.923

Dependent variable: healthy diet as the reference category.

**Table 7 nutrients-15-03591-t007:** Comparison of adherence to a healthy diet with lifestyle habits and frequency of junk food consumption with the test of equality for column proportions (z-test).

Lifestyle Habits	Adherence to Healthy Diet					
	Unhealthy Diet (a)		Medium Healthy Diet (b)		Healthy Diet (c)	
	n	%	n	%	n	%
Total	438	25.48	960	55.85	321	18.67
Exercise frequency (χ^2^ = 54.58, *p* < 0.001)						
Not	136 ^b,c^	31.05	152 ^c^	15.83	27	8.41
Yes, very rarely	196 ^c^	44.75	402 ^c^	41.88	92	28.66
Yes, 2–3 times a week	64	14.61	237 ^a^	24.69	104 ^a,b^	32.40
Yes, every day under an hour	22	5.02	90 ^a^	9.38	36 ^a^	11.21
Yes, daily for at least an hour	20	4.57	79 ^a^	8.23	62 ^a,b^	19.31
Smoking (χ^2^ = 71.29, *p* < 0.001)						
Yes, excessive daily	134 ^b,c^	30.59	163 ^c^	16.98	34	10.59
Yes, 1–2 cigarettes daily	34	7.76	63	6.56	17	5.30
Yes, 2–3 times a week	11	2.51	16	1.67	4	1.25
Yes, occasionally	49 ^c^	11.19	90 ^c^	9.38	18	5.61
Not	210	47.97	628 ^a^	65.42	248 ^a,b^	77.26
Sleep time, hours (χ^2^ = 54.64, *p* < 0.001)						
I have frequent insomnia	41	9.36	60	6.25	19	5.92
Under 7 h per night	188 ^b,c^	42.92	276	28.75	87	27.10
Over 9 h a night	24	5.48	34	3.54	8	2.49
7–8 h per night	185	42.24	590 ^a^	61.46	207 ^a^	64.49
Frequency of junk food consumption (χ^2^ = 49.29, *p* < 0.001)						
Very rarely or not at all	38	8.68	231 ^a^	24.06	160 ^a,b^	40.84
2–3 times a month	76	17.35	255 ^a^	26.56	83 ^a^	25.86
2–3 times week	170 ^b,c^	38.81	217 ^c^	22.60	32	9.97
Once a week	94 ^c^	21.46	226 ^c^	23.54	44	13.71
Daily	60 ^b,c^	13.70	31 ^c^	3.23	2	0.62

Values in the same row not sharing the same superscript were significantly different at *p* < 0.05 in the two-sided test of equality for column proportions (z-test). Tests were adjusted for all pairwise comparisons within a row of each innermost sub-table using the Bonferroni correction.

**Table 8 nutrients-15-03591-t008:** Results of the multinomial logistic regression for adherence to healthy diet.

Independent Variables	Unhealthy Diet			Medium Healthy Diet		
	OR	95% CI	*p*	OR	95% CI	*p*
Exercise frequency						
Not	1			1		
Yes, very rarely	0.423	(0.261–0.685)	<0.001	0.776	(0.486–1.239)	0.776
Yes, 2–3 times a week	0.122	(0.073–0.205)	<0.001	0.405	(0.253–0.648)	<0.001
Yes, every day under an hour	0.121	(0.062–0.238)	<0.001	0.444	(0.253–0.780)	0.005
Yes, daily for at least an hour	0.064	(0.033–0.123)	<0.001	0.226	(0.134–0.384)	<0.001
Smoking						
Yes, excessive daily	4.654	(3.062–7.076)	<0.001	1.893	(1.272–2.818)	0.002
Yes, 1–2 cigarettes daily	3.215	(1.817–5.688)	<0.001	1.463	(0.840–2.550)	0.179
Yes, 2–3 times a week	3.248	(1.019–6.350)	<0.001	1.975	(1.166–3.344)	0.011
Yes, occasionally	2.362	(1.283–4.349)	0.006	1.580	(0.523–4.771)	0.418
Not	Reference					
Sleep time, hours						
I have frequent insomnia	0.999	(0.548–1.820)	0.996	0.995	(0.563–1.759)	0.995
Under 7 h per night	1					
Over 9 h a night	1.388	(0.600–3.214)	0.444	1.340	0.598–3.002	0.478
7–8 h per night	0.414	(0.300–0.571)	<0.001	0.898	(0.673–1.199)	0.467
Frequency of junk food consumption						
Very rarely or not at all	1			1		
2–3 times a month	3.855	(2.407–6.175)	<0.001	2.128	(1.546–2.929)	<0.001
2–3 times week	5.368	(3.333–7.528)	<0.001	4.697	(3.079–7.165)	<0.001
Once a week	4.995	(2.438–7.879)	<0.001	3.558	(2.431–5.206)	<0.001
Daily	8.316	(3.555–13.865)	<0.001	7.736	(2.533–12.497)	0.002

## Data Availability

There are no data available for this publication.

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
