# Peer review of "Evaluation of Junk Food Consumption and the Risk Related to Consumer Health among the Romanian Population"

_nutrients, 2023, doi:10.3390/nu15163591_

Round 1

Reviewer 1 Report

1.The main question addressed by this study is to know the characteristics of junk food in the Romanian population.

2.The topic is quite original and relevant since junk food is becoming more and more prevalent in most countries of the world, replacing the traditional and generally healthier diet.

3.This study provides relevant information on junk food in a specific country, which can be used to establish comparisons.

4.It helps to know the current situation and the reality of a specific country regarding junk food.

5.I believe that there is no need to include anything in the methodology as all the necessary aspects have been addressed.

6.The conclusions show great coherence and relate very well to the objectives and the main question of the study.

7.The references are very well matched and very current (low obsolescence rate).

8.Tables and figures facilitate reading and help to better understand the study. They are a complement to the text.

Author Response

Thank you for the effort put into evaluating the manuscript and for appreciating it.

Reviewer 2 Report

1.The logical reasoning and cohesion of the introduction is not very strong, please revise it if possible.   2. Discussion can add other  research for comparative analysis.  3. Conclusion  does not highlight the innovative findings of this study well.

Author Response

Thank you for your valuable comments. We took into account all the indications and we made the following additions:

Introduction

The World Health Organization (WHO) draws attention to the danger of excessive consumption of unhealthy foods and drinks, which represent an important risk factor for non-communicable diseases (NCDs) [1]. Junk food, sweetened and alcoholic drinks are included in the category of foods with a major risk for metabolic syndrome complicated with serious consequences for health and even for the risk of premature death in the case of long-term excessive consumption associated with an unbalanced and sedentary life-style. These food products are characterized by hypercaloric content, low in nutrients and rich in additives [2-6].

Discussion

Junk food products are rich in additives, some of which (such as food sweeteners, flavorings, monosodium glutamate, etc.) play an important role in the development of food addictions [48-50].

Unfortunately, the consumption of junk food products is usually accompanied by the consumption of sweetened beverages and even smoking or alcoholic beverages (which adds a series of dangerous compounds for the body and a significant increase in calories) and less vegetable products which would ensure an intake of important fibers and antioxidants for the detoxification of the body [57-59]. As a result, in the long term, unhealthy and unbalanced food associations are also the basis of micronutrient malnutrition, strongly altering the state of health [60].

Conclusions

The consumption of junk food becomes dangerous for health when it is accompanied by a sedentary lifestyle and is especially associated with other hypercaloric foods such as sweetened or alcoholic beverages. Physical activity helps to consume excess calories and the regular consumption of vegetable and fruit products counterbalances the absence or deficiency of important nutrients for the body such as fibers, antioxidants, vitamins, minerals, unsaturated fats.

Thank you for the effort and for the help provided to improve the manuscript.

Thank you for your valuable comments. We took into account all the indications and we made the following additions:

Introduction

The World Health Organization (WHO) draws attention to the danger of excessive consumption of unhealthy foods and drinks, which represent an important risk factor for non-communicable diseases (NCDs) [1]. Junk food, sweetened and alcoholic drinks are included in the category of foods with a major risk for metabolic syndrome complicated with serious consequences for health and even for the risk of premature death in the case of long-term excessive consumption associated with an unbalanced and sedentary life-style. These food products are characterized by hypercaloric content, low in nutrients and rich in additives [2-6].

Discussion

Junk food products are rich in additives, some of which (such as food sweeteners, flavorings, monosodium glutamate, etc.) play an important role in the development of food addictions [48-50].

Unfortunately, the consumption of junk food products is usually accompanied by the consumption of sweetened beverages and even smoking or alcoholic beverages (which adds a series of dangerous compounds for the body and a significant increase in calories) and less vegetable products which would ensure an intake of important fibers and antioxidants for the detoxification of the body [57-59]. As a result, in the long term, unhealthy and unbalanced food associations are also the basis of micronutrient malnutrition, strongly altering the state of health [60].

Conclusions

The consumption of junk food becomes dangerous for health when it is accompanied by a sedentary lifestyle and is especially associated with other hypercaloric foods such as sweetened or alcoholic beverages. Physical activity helps to consume excess calories and the regular consumption of vegetable and fruit products counterbalances the absence or deficiency of important nutrients for the body such as fibers, antioxidants, vitamins, minerals, unsaturated fats.

Thank you for the effort and for the help provided to improve the manuscript.